# Factors, Barriers, and Recommendations Related to Mobile Health Acceptance among the Elderly in Saudi Arabia: A Qualitative Study

**DOI:** 10.3390/healthcare11233024

**Published:** 2023-11-23

**Authors:** Jwaher A. Almulhem

**Affiliations:** Medical Informatics & e-Learning Unit, Medical Education Department, College of Medicine, King Saud University, Riyadh 11461, Saudi Arabia; jalmulhem@ksu.edu.sa

**Keywords:** mHealth, intention to use, digital health, older adult, H-TAM model, Saudi Arabia

## Abstract

The use of mobile health (mHealth) is providing value to the elderly, but their acceptance of it is lower than in other age groups. Thus, this study aims to explore the factors influencing their intention to use mHealth and identify barriers and suggested solutions among elderly people aged 50+ years in Saudi Arabia, guided by the Healthcare Technology Acceptance Model (H-TAM). In this qualitative study, 14 elderly people (six females and eight males) were recruited. Participants were included if they were Saudi, aged 50+ years, and used smartphones. Participants were engaged in semi-structured interviews, which were transcribed verbatim and thematically analyzed. Peer review was conducted and saturation was reached to maintain rigor. Three major themes emerged: (1) factors affecting intention to use, (2) concerns and barriers, and (3) solutions and recommendations. Influenced factors were identified as perceived usefulness, perceived need, perceived ease of use, perceived benefits, familiarity, trust in technology, advice acceptance, facilitating conditions (family support), and compatibility. Older participants, particularly those with lower educational attainment, displayed less familiarity with mHealth. Lack of digital literacy, health and aging issues, worry about making mistakes, and social issues emerged as central barriers. Addressing these factors in the design and promotion of mHealth can enhance its successful adoption among the elderly.

## 1. Introduction

Population aging is a pressing issue in different countries around the world [1]. The number of Saudi aged 60 years and older is 1.36 million, which represents 3.81% of the population. This number is expected to increase to approximately 2.1 billion by 2050 [2]. This increase in the size of the aging population is positively correlated with the increasing prevalence of chronic diseases and multi-morbidity, such as diabetes and hypertension [3]. Consequently, emerging challenges may face the healthcare system [4]. 

The use of mobile health (mHealth) has added value to older adults in terms of cost savings, improving health providers’ engagement, and reducing caregiver stress [5]. MHealth is defined as “medical and public health practice supported by mobile devices” [6]. On the other hand, the elderly population has faced many barriers and challenges during the use of mHealth that might negatively affect their experience [7]. Furthermore, they may face barriers that are different from barriers affecting other age groups. Prior studies have reported several concerns and barriers, including ease of use, cost, digital literacy, trust in technology, patients’ attitudes, and physical and cognitive challenges [7,8].

Several models have been introduced to help understand the factors that have influenced the acceptance of technology among users. One of the fundamental models is the Technology Acceptance Model (TAM) which identifies perceived usefulness, perceived ease of use, and social norms as factors [9]. The Unified Theory of Acceptance and Use of Technology (UTAUT) and its extension focuses on performance and effort expectancies, social influence, hedonic motivation, price value, and habit [10,11]. A model that concentrates on the elderly population’s technology acceptance is the Senior Technology Acceptance Model (STAM) [12]. However, these models have only studied the acceptance of technology in different fields with a limited concentration on healthcare. The Healthcare Technology Acceptance Model (H-TAM) for older adults was introduced to illustrate factors that influence the acceptance of healthcare technology among hypertensive older adults. The model identified 15 factors; 13 factors were included in other models and two factors were novel in the model [13].

Prior studies discussed the intention to use mHealth among different populations. A recent cross-sectional study conducted among Dutch elderly found that men had higher intentions to use mHealth than women. Perceived usefulness, perceived ease of use, and attitude toward use were factors affecting the intention to use mHealth. The study concentrated more on gender differences and missed studying other demographic factors. Furthermore, this study was based on the TAM model, which was not specifically intended for a healthcare context [14]. Another quantitative convenience sample study preformed among Shanghai elderly concluded that familiarity, usage of mobile health, self-rated health, perceived ease of use, perceived usefulness, and self-efficacy were important factors influencing the willingness to use mHealth [15]. A quantitative study conducted in Bangladesh indicated that social influence, price value, habit, and service quality significantly impacted the intention to adopt mHealth [16]. However, this study was based on the UTAUT model, which is not focused on a healthcare environment.

In the Kingdom of Saudi Arabia (KSA), many mHealth technologies have been widely introduced in response to the COVID-19 pandemic, such as Sehha and Wasfaty [17]. On top of that, the population will have unified digital medical records by 2025 which will enable them, including the elderly population, to access their health information [18]. Prior research has examined factors that can influence the acceptance and use of an electronic personal health record among the general population. The study defined performance expectancy, effort expectancy, social influence, and technological literacy as determinants [19]. Other studies focused on the use of digital health among healthcare professionals and medical students [20]. However, no studies have qualitatively discussed factors influencing the intention to use mHealth among the elderly in KSA. The findings from such a study can help inform the design and implementation of mHealth technologies, specifically for the elderly who may be less comfortable with technology use.

Therefore, the objective of this qualitative study is to explore the factors influencing intentions to accept or not accept the use of mHealth, and to identify barriers and suggested solutions among the elderly aged 50+ years in KSA guided by H-TAM.

## 2. Materials and Methods

### 2.1. Participants

In this qualitative study, purposeful sampling was conducted through a network of researcher contacts and several retirement associations to recruit the participants [21]. A sample of 14 older adults aged 50 years and older (six females and eight males) was recruited. Participants were included in the study if they were Saudi, over the age of 50 years, and used a smartphone. They were excluded if they were unable to consent independently. Invitation letters were distributed to interested individuals which included the objective of the study and the duration of the interview. Once eligibility had been confirmed, participants were asked to provide informed consent, before being interviewed, for us to audio-record the interview, use anonymous quotations, and conduct the interview in Arabic, since it is the primary language of the participants. Until thematic saturation was achieved, the primary researcher recruited new participants and conducted interviews. The institutional review board (IRB) at the College of Medicine, King Saud University reviewed and approved this study.

### 2.2. Healthcare Technology Acceptance Model

The intention to use factors that were used in this study was derived from the H-TAM model. The model was recently developed based on reviewing several technology acceptance models and concentrated on the elderly and healthcare technology context [13]. It introduced familiarity, perceived need, perceived benefits, advice acceptance, and trust as new factors that have not previously emerged in technology acceptance models. This model suggests that intention to use is based on four main factors, including perceived usefulness, perceived ease of use, facilitating conditions, and social influence [13]. Perceived usefulness is related to the degree to which an individual thinks that using the selected technology would enhance his/her quality of life [11]. Perceived need, perceived benefits, and relative advantages are considered as factors related to perceived usefulness [13]. Venkatesh et al. described perceived ease of use as “the degree of ease associated with the use of the system” [11] (p. 450). According to the model, perceived ease of use is influenced by complexity and familiarity, which are related to privacy and trust in technology [13]. Facilitating conditions are described as any conditions available that support use of technology according to STAM [12], which is related to price value and perceived compatibility. Social influence focuses on the perception an individual has regarding the importance of others’ belief that he or she should use the technology [11]. It hypothesizes that this is associated to advice acceptance and trust in a person [13]. A complete description for each H-TAM variable can be found in the study by Harris [22].

### 2.3. Data Collection

A semi-structured interview guide (Appendix A) was designed based on H-TAM [13] and extensive research on the literature in order to identify other barriers related to the intention to use mHealth found among the elderly. In developing the interview guide, we followed the framework suggested by Kallio et al. [23]. The interview guide was reviewed by an expert in the field and necessary changes were made to come up with the final interview guide. The primary researcher conducted the interviews either face-to-face or by telephone based on the participant’s preference. Prior to each interview, the researcher explained the research purpose, design, and interview questions in detail to each participant. During the interview, an example of mHealth available in KSA, known as the Sehhaty Platform, was also discussed with each participant. This is the unified platform of the Ministry of Health, which allows citizens and residents to access health information and benefits from several health services offered by various entities in the healthcare sector for free. The platform offers several services, such as immediate virtual consultations, appointment booking, medical test results, and counting steps [24]. The interview lasted between 20 and 50 min. The primary researcher transcribed verbatim all the audio recordings. Peer review was conducted by an expert in qualitative research to ensure the study’s rigor [25].

### 2.4. Data Analysis

To analyze the responses of semi-structured interviews, thematic content analysis was deployed as described by Braun and Clarke [26]. The analysis used both theory-driven and data-driven approaches. A qualitative software program, Dedoose version 9.0.46, was used for the coding and analysis of the transcripts [27]. First, the primary researcher translated the interview scripts from Arabic to English and read all the transcripts to build initial understanding. Secondly, codes and sub-themes were created based on participants’ responses. To generate themes, codes and sub-themes were reviewed deeply. Subsequently, all quotations were identified and linked to sub-themes. All quotations were added to the thematic analysis document. During analysis, the researchers combined data using an inductive and theory-based approach. Specifically, their conclusions were taken from the literature in terms of factors identified in H-TAM and other models. The primary researcher and an expert in qualitative studies conducted the analysis together to avoid misinterpretations.

## 3. Results

### 3.1. Participant Description

The study comprised 14 participants (Table 1), most of them married and aged above 60 years old. The participants included six females and eight males. They were all smartphone and internet users with varying educational backgrounds. Most of the participants had income levels ranging from 0 to 10,000 SR per month. Regarding self-rated health, the majority of participants rated their health as “Good”, indicating generally good health among the interviewees. Most participants had chronic diseases such as diabetes, hypertension, or high cholesterol levels.

### 3.2. Thematic Analysis

From the interview data, we assigned 81 codes and created three major themes: (1) factors affecting intention to use mHealth, (2) concerns and barriers of intention to use, and (3) solutions and recommendations.

#### 3.2.1. Theme 1: Factors Affecting Intention to Use mHealth

We derived six subthemes that reflect factors that affect the intention to use mHealth based on the H-TAM model (Table 2).

Sub-theme 1: role of demographics and health status

Demographics and health status reveal multifaceted factors influencing the intention to use mHealth among participants. Age, gender, and level of education were influential factors in the intention to use mHealth. Generally, younger individuals exhibited a greater sense of ease with technology, enabling them to easily adapt to and utilize it. One participant explained, “*It depends on the fact that a person has experienced it from the beginning of his life, but if his interest in it is late because a person’s skills weaken with age*”. Conversely, older participants, particularly those with lower educational attainment, displayed less familiarity with technology and voiced concerns about its use. One interviewee mentioned, “*The elderly do not use technology…. refuse to learn*”. However, another participant highlighted, “*The more educated an older person is, the easier the technology will be for him*”. This sub-theme acknowledges the potential barriers faced by older adults who may not have had the same educational opportunities. Three out of six female participants perceived mHealth technology as not useful, as mentioned by one participant: “*I prefer to go to the hospital and book appointments*”. Most of the participants who think mHeath is not easy to use and have negative attitudes are female.

Furthermore, the intention to use mHealth was influenced by the participants’ health status. Participants who were more attentive to their health or had ongoing medical conditions demonstrated a higher propensity to employ mHealth for the purposes of monitoring their well-being, scheduling appointments, and seeking medical guidance. As one interviewee with diabetes noted, “*I have diabetes, and I use the app regularly to keep track of my glucose levels and medications. It’s become an essential tool for managing my condition*”. Nonetheless, even individuals in good physical condition acknowledged the advantageous nature of these technologies in terms of booking appointments and attaining healthcare services in an efficient manner. Another participant, despite being generally healthy, emphasized, “*Even though I’m generally healthy, I like having technology because it makes booking doctor’s appointments so much easier. It saves me a lot of time and hassle*”. These quotes illustrate how health status plays a pivotal role in shaping the intention to use mHealth among older adults.

Sub-theme 2: perceived usefulness

Perceived usefulness: the thematic analysis revealed a strong agreement among participants regarding the perceived usefulness of mHealth. Many participants expressed that these technologies serve as valuable tools for health management. As one interviewee remarked, “*Anyone who cares about their health would find this technology beneficial*”. Participants also highlighted the reason for this agreement: “*Yes, it is very useful because health awareness has become widespread among the elderly*”. These perceptions collectively indicate a positive outlook on the utility of mHealth. However, a few participants disagree with the perceived usefulness of mHealth in some situations, as demonstrated by the following response: “*…but if the patient is in an emergency situation, this technology does not help the patient*”.

Perceived benefit: the discussions emphasized the perceived benefits of mHealth, which were often intertwined with their usefulness. Participants expressed that mHealth effectively enhances health management and facilitates easy access to healthcare services. As one participant mentioned, “*An elderly person may not have a suitable means of transportation to transport him, or he may be disabled and unable to move, so this app and technology will help him a lot*”. The perceived benefits include improved accessibility to clinical services and patient data, streamlined processes, and reduced waiting time: “*It saves us time, we reach the goal quickly, and it is easy to choose the service. These are incentives to continue using mHealth applications and technology*”.

Perceived need: most participants indicated their immediate need for mHealth to help them with appointment and vaccination booking and medication management. As one interviewee remarked, “*Yes, it is useful to know the appointments and timing of medications. As well as medication prescriptions. What I need most is to book appointments*”. Some participants who had chronic conditions, such as hypertension, emphasized the necessity of such technology in managing their health: “*Yes, I [will] use it for alerts for treating high blood pressure, as well as calculating the number of steps and appointments*”. Some participants questioned the perceived need for others. One participant pointed out, “*The elderly person wants to meet the doctor face to face, especially if he is not accustomed to it. Those who have tried such technology among the elderly are a small group of elderly people, only possibly educated people*”. Other participants agreed with the need for mHealth, even for other elderly people, particularly for those who have the ability to use such technology: “*It is useful for people who know how to use it because it is necessary*”. This suggests that the perceived need for mHealth can be contingent on an individual’s health status and ability to use it.

Relative advantage: most of the participants have not used any mHealth technology that could be compared to the Sehhaty app. As one of the participants mentioned, for example, “*No, I don’t have any similar technology*”. Except for three participants, many stressed the importance of availability of healthcare services through a mHealth app in order to ensure its usefulness: “*But it [ the other app] is difficult to book appointments because they are only available within 3 months*”.

Sub-theme 3: ease of use

Perceived ease of use: most participants expressed degrees of ease in using mHealth. As one participant stated, “*….of course, now all programs and applications are easy, you just need to enter through your mobile phone*”. The participants acknowledged mHealth technology’s perceived ease of use compared to traditional methods of providing care and linked it with benefits provided: “*There is a big difference between the traditional method and using such technology. Using it reduces congestion and regulates the time of patient entry into health centers*”. However, some participants indicated that mHealth technology’s ease of use depends on several factors: “*…but if he is uneducated, uncultured, and does not care about his health, it will be difficult for him*”. This suggests that the perceived ease of use may not be uniform and can depend on individual familiarity and digital literacy.

Convenience/inconvenience: convenience emerged as a significant factor influencing the intention to use mHealth. For some, there was an acknowledgment of the effort required to learn how to use mHealth: “*Of course there is effort, the elderly need to learn how use this technology*”. However, other participants did not believe in making any effort while using mHealth: “*No, there is no effort or anything*”. One participant explained the main reason for putting in effort when using mHealth as follows: “*For an elderly person who is proficient in skill and uses technology, there will not be an effort. But for those who do not use technology, it will be difficult for them*”.

Familiarity: familiarity with technology was a recurring theme in the interviews. Most participants are familiar with mHealth, as one participant mentioned, “*Average what I consider good. I can book appointments for example, I mean just making choices*”. A few participants who are well-educated described their familiarity with mHealth as “*Excellent*”. However, few participants expressed their challenges in using mHealth. This might imply that familiarity with mHealth technology is linked with education level.

Privacy and trust: most of the participants did not see privacy as an issue or concern while using mHealth. As one participant stated, “*I do not think it affects patient privacy. Because the patient wants the doctor to know everything about his health condition*”. Another participant highlighted the impact of social communication on increasing privacy concerns among the elderly: “*Only if the topic becomes common among people in their daily conversations can an elderly person begin to fear the issue of privacy. But does he really know if his health status has been authorized or tampered with? I don’t think so*”. Only a few participants expressed their concern regarding privacy and indicated that the elderly are considered an easy target for illegal users. As one participant stated, “*The topic is sensitive. No one wants to reveal that he has such-and-such a disease because they do not want anyone to know about the disease, or their medical history, as it is confidential information*”. They acknowledged the existence of several security measures for data in such technology. For example, one interviewee mentioned, “…*do not share the password except with those you trust and with your treating doctor, because they are private information about you*”. This sub-theme underscores the significance of data security awareness in the adoption of mHealth.

Participants expressed a high level of trust in mHealth and linked it with benefits provided and increased awareness among the elderly. One participant emphasized, “*My trust in this technology is great, and it may reach the point of blind trust. I have benefited from their advantages, saved me time, and saved me effort*”. Some participants highlighted the role of the technology’s developers in building trust “*Because those responsible for it are trustworthy people*”. Only one participant has a lack of trust in mHealth and preferred to use traditional methods of providing care due to the absence of perceived ease of use. This highlights the pivotal role trust plays in the acceptance of mHealth.

Sub-theme 4: role of social factors

Subjective norm: most of the participants were not clear about the role healthcare providers play in shaping the subjective norms of older adults “*It is normal and the doctor will not ask about it*”. Except for four participants who expressed positive feelings about the role of healthcare providers, as one participant shared “*It feels excellent. The doctor will see that I am a health-conscious person*”.

The influence of family members and friends was positive in participants’ perceptions, particularly when they shared using the same app. One interviewee remarked, “*My family is also happy that I use this technology and rely on myself*”. Additionally, another interviewee stated, “*I did not share with my friends that I use such technology, but I expect that they will be happy*”. However, only a few participants did not care about their family and friends’ roles in mHealth technology’s intention to use: “*This is up to me. There is no one who encouraged me or discouraged me from using technology*”. These comments underscore the impact of family and friends’ dynamics on the subjective norm.

Advice acceptance: the acceptance of advice from healthcare providers emerged as a factor. When participants received recommendations from their healthcare providers, it positively influenced their attitudes and confidence in using mHealth. As one interviewee expressed, “*Yes, if the doctor explains the service, I will surly use it*”. Two participants highlighted that their acceptance will be based on their need and the benefits of the new mHealth technology “*…but in the event that the app will increase my burden or that I will not need it and it benefits another group, no, I will not use it*”.

The acceptance of advice from family members and friends played a role in participants’ willingness to use mHealth. One participant stated, “*Yes, if a member of my family suggests it to me, I will trust the technology*”. Only a few participants were not willing to take advice from their friends. One interviewee mentioned, “*I do not trust people from outside my family*”. This indicates the importance of both healthcare providers and family as trustful sources related to the adoption of mHealth.

Sub-theme 5: facilitating conditions

Compatibility: participants highlighted the integration of mHealth during their daily routine as a significant feature affecting their intention to use it. Most of the participants revealed that mHealth is compatible, particularly participants with chronic diseases: “*Yes, because the elderly person tries to know a lot about his health, the number of steps he takes, and the effect of his medications on his health, and this information is available in the application. For example, a diabetic patient can know the impact of steps on his cumulative blood sugar level*”. Other participants emphasized the lack of need to integrate mHealth into their routine since it might increase their anxiety and negatively affect stability. One participant illustrated, “*I don’t expect to need to incorporate it into my daily routine. There is no need for me to stress myself if my blood sugar increases*”. This emphasizes the importance of the type of information presented in mHealth and used by the elderly.

Price value: participants were evenly distributed between willing to pay and not to pay for mHealth technology. Participants expressed concerns about the cost associated with mHealth. One interviewee noted, “*If it was a small amount, for example, 100 riyals, I could pay it*”. Another linked the cost with the benefits provided by mHealth, saying, “*Yes, depending on the benefit it provides to me. The lower the amount, the better. The annual subscription can be, for example, 20 riyals per month*”. On the other hand, other participants were not willing to pay for mHealth and linked it with a lack of perceived need for such technology: “*But if they ask me to pay for technology, I will not pay. Because I won’t need it*”. This highlights the effect of the perceived cost of mHealth on the intention to use it.

Support availability: the role of the family in supporting older adults in using mHealth was clear among our participants, especially if there is a lack of familiarity with using such technology. One interviewee mentioned, “*It is possible for him to turn to his daughters or sons who know how to use the technology*”. Some participants preferred to use mHealth by themselves since they already have the skills required to use them: “*I don’t need help and I feel comfortable. Because I am proficient in using it*”. These insights underline the importance of familiarity and the role of the family in bridging the familiarity gap.

Sub-theme 6: attitude

Among the participants, the sub-theme of the attitudes of older individuals emerged. Most participants demonstrated a positive attitude and interest in using mHealth technology, particularly if they already had the ability to use such technology. For example, one interviewee mentioned, “*If I am able to use the technology, I will feel familiar and knowledgeable*”. Participants who had negative attitudes regarding mHealth expressed their dislike of using mHealth due to struggles with using it: “*But I want to go to the hospital, it is better for me. Because I don’t like using the mobile phone a lot, only messages or calls*”. This highlights the importance of addressing digital literacy concerns among older adults.

#### 3.2.2. Theme 2: Concerns and Barriers

Central barriers identified by our participants were categorized into digital literacy, technical support, worry about mistakes, health, and social barriers (Figure 1). The top issues mentioned were related to a lack of digital literacy, which negatively impacts the intention to use mHealth effectively. As one participant stated, “*Main barrier if the elderly does not know how to use the device*”. This underscores the apprehension some older adults may have regarding their technological proficiency and the challenges they anticipate. Some participants expressed concerns about health and age-related issues that prevent them from using mHealth technology: “*Health challenges, such as poor eyesight, may be the reason that prevents you from using the technology*”. Another participant expressed, sadly, an inability to remember: “*If I practiced it, I might forget how to use it later*”. Social and cultural factors related to convincing the elderly about mHealth emerged as a barrier. One interviewee highlighted, “*The problem is in culture and the culture of ignorance. It is possible that the elderly person is ignorant and refuses to learn*”. Fearing to make mistakes while using mHealth technology was stated as a barrier, particularly among uneducated elderly, as mentioned by one participant: “*…it is that part of them is afraid of making mistakes when they use it*”. The last barrier mentioned was related to the availability of internet connection and appropriate devices. As one participant stated, “*the lack of availability of the device, and this leads to meeting the doctor in person, as well as the lack of Internet in the desert areas*”.

#### 3.2.3. Theme 3: Solutions and Recommendations

Several key solutions were identified by participants in order to address these barriers and challenges and encourage the intention to use mHealth among older adults (Figure 2).

Solution 1: education and awareness. Many interviewees highlighted the importance of education and awareness as a foundation for successful mHealth adoption. “*Urging individuals to teach the elderly and make them aware of the importance of technology and provide support*”, noted one interviewee. To address this, comprehensive educational campaigns should be launched to inform older adults about the benefits, functionalities, and security measures associated with mHealth. These campaigns should be conducted through multiple channels, including healthcare providers, community centers, and media. One participant mentioned, “*It must have publicity in the hospital to encourage the elderly to use it. It makes him excited to learn and use it*”. Training and education should be also provided to healthcare providers who will provide medical services to the elderly through using mHealth, as one participant indicated: “*It depends on the service provider. If he has the ability, mechanism, and correct information to deal with this category of society, the technology will be successful*”.

Solution 2: user-centric design. One of the recurring points expressed by older adults was the difficulty of using mHealth: “*They should make it easier for the elderly*”. To address this, developers should focus on creating intuitive and simplified user interfaces. One interviewee suggested, “*There will be a special version for those who have reached the age of 60*”. Another participant focused on using similar technological interfaces that the elderly are already familiar with: “*…communicating with patients should be using the same technology that the elderly use like WhatsApp*”.

Solution 3: motivation and engagement: integrating some elements into mHealth can make them more engaging and motivating. “*Any technology that contains feedback on communication and service provision is considered excellent and attracts the user to use it*”, one participant expressed. Providing guidance messages to the elderly will demonstrate the importance of mHealth, as mentioned by one interviewee: “*It is important that such technology gives guidance messages to the elderly provided by the Ministry of Health*”.

Solution 4: continuous practice. Several participants highlighted the importance of practicing the use of mHealth technology: “*The elderly learn this technology and try them more than once until they master their use and then use them*. *The use will be better*”. Another participant said, “*at first it will be difficult, but if you get used to it, it will become easier*”, indicating the relationship between continuous practice and perceived ease of use.

## 4. Discussion

The adoption of mHealth, particularly among older adults, is indeed a heterogeneous process shaped by a multitude of factors [28]. To gain a more comprehensive understanding of this phenomenon, this qualitative study used the H-TAM model to serve as a valuable framework focused on insights from participants to discuss intention to use the Sehhaty App v3.7.1, as an example of mHealth technology. Indeed, the Sehhaty App is one of the most downloaded government mHealth technology apps provided in KSA [29]. H-TAM provides a structured framework for comprehending how the elderly embrace and utilize healthcare technology [13]. Indeed, it has been suggested that further research is required to link intention to use factors and the existing technology acceptance model [30]. When examining the interview responses within the context of H-TAM, we discerned the critical elements that influence the intention to use mHealth technology among older individuals in KSA. Perceived usefulness, perceived need for mHealth technologies, perceived ease of use, perceived benefits, familiarity, trust in technology, advice acceptance, facilitating conditions (family support), and compatibility were identified as influential factors. Age, gender, and level of education were recognized as moderators that impact the intention to use mHealth. Lack of digital literacy, health and aging issues, worry about making mistakes, and social issues emerged as central barriers that affect intention to use mHealth technology among the elderly. Tailored educational and training campaigns for both the elderly and healthcare providers are important. User-friendly interfaces and simplicity in design can mitigate these barriers.

Numerous interviewees acknowledged that the perceived usefulness has an influence on the intention to use mHealth. This finding was similar to several studies conducted among the elderly in Bangladesh, Holland, and China [14,15,31]. Perceived benefits are directly linked to perceived usefulness in the H-TAM model [13]. Indeed, the findings of this study indicate that the Sehhaty app provides several perceived benefits for the elderly in terms of managing health data, saving time and effort, and streamlining clinical services. These perceived benefits have a positive impact on elderly mHealth acceptance, particularly in the pre-implementation phase [30]. The second factor that helped to define perceived usefulness was the perceived need. Perceived needs are defined as whether or not the elderly person believes that he/she “needs” the technology currently [13]. The Sehhaty app provides a wide range of services that meet the needs of the elderly, including immediate virtual consultations and appointment booking, as well as access to health information [24]. According to systemic review, the presence of perceived need increases the likelihood of acceptance of technology among the elderly [30].

Interviewees expressed varied perceptions of the ease of using the Sehhaty app. Some elderly people found it straightforward to set up profiles, book appointments, and access health information. However, other participants encountered challenges due to differences in their backgrounds, education, and lack of familiarity and digital literacy. According to a recent review, the most important facilitator for older people using technology is that technology should be easy to use [32]. Similarly, a recent study conducted among Dutch elderly found that perceived ease of use was a factor affecting intention to use mHealth [14]. According to the H-TAM model, perceived ease of use is directly related to trust and privacy, which are linked to familiarity [13]. Trust is a crucial element in the adoption of mHealth. Trust encompasses confidence in the technology’s security, reliability, and the credibility of information provided. The Sehhaty app, being a government-backed platform, instills a high level of trust among users. Indeed, privacy was not defined as a concern among participants since the level of trust in technology is high among them. Familiarity with mHealth was clear among participants and acknowledged as an influence on the intention to use it, which is similar to other studies [15,30,31]. Designing user-friendly interfaces and providing comprehensive tutorials can address these concerns and enhance perceived ease of use. Indeed, poor screen design in terms of small font, small buttons, and small space between buttons were recognized as features that participants disliked when they used a specific diabetes app [7]. Furthermore, there is a need to tailor mHealth to accommodate different age groups and provide continuous support to mitigate age-related barriers and enhance digital literacy among the elderly.

Social influence, including advice acceptance, is a unique factor in the H-TAM model. Interviewees acknowledged the role of social influence in their decision to adopt mHealth technology, which has been established in prior studies [30,31]. However, Maswadi et.al. found that social influence has no significance on the intention to use smart home among the elderly in KSA [33]. Indeed, accepting advice from healthcare providers and families is more relevant to health-related behavior changes, and it is a sensitive matter [13]. The responses underscore the significance of advice acceptance and recommendations from trusted sources to overcome barriers related to convincing the elderly to use mHealth technology, which was discussed in relation to social and cultural barriers identified in the study. Healthcare providers, family members, and friends play a pivotal role as facilitators in encouraging older adults to embrace mHealth. Endorsements and guidance from these sources enhance the perceived credibility of mHealth.

This study found out that facilitating conditions in terms of the presence of family support is an influencing factor in enhancing intention to use mHealth technology, which is in alignment with H-TAM [13]. The interview responses revealed varying levels of facilitating conditions. While some older adults possess the skills and resources to adopt mHealth technology independently, others still rely on external assistance. This finding is similar to another review study which indicated that both social support and lack of social support were reported as facilitators [32]. Having family support for the elderly is necessary to overcome the digital literacy gap [32]. However, elderly people who have the required technical skills rely on themselves to use mHealth technology. Indeed, this finding indicates the significant role of families’ support which acts as a trusted source in bridging the digital literacy gap among some elderly which was identified as a central barrier in the study. More importantly, this finding fills the gap in the literature related to the type of social support needed to enhance the intention to use mHealth among the elderly [32].

Interviewees shared their perceived compatibility with mHealth technology as a factor that encourages their intention to use mHealth technology. They want the integration of mHealth activities into their existing routines. Moreover, features such as rewards, challenges, and progress tracking can also encourage older adults. Indeed, goal setting features were one of the most favored features selected by older participants who used diabetes self-management apps [7]. Gamification elements, social connections within technology, and the integration of mHealth activities into daily routines can fuel motivation and sustained engagement with technology.

In terms of mediators, this study recognized age, gender, and level of education as moderators that impact the intention to use mHealth. Prior studies have found a lower acceptance of mHealth technologies among elderly users compared to users in other age groups [34]. Similarly, the use of the internet among the elderly in Saudi Arabia has declined with age. It reached 74% among elderly aged more than 70 years old compared to 93% among elderly aged between 60 and 65 years old [29]. Indeed, the elderly are a highly heterogenous age group with differing needs and they require specific mHealth solutions [35]. Health and aging issues, such as lack of memory and poor eyesight, were identified as barriers in this study. Focusing on user-centric design that meets the special needs of the elderly is required [36]. Level of education plays a significant role in digital literacy and familiarity with mHealth. This finding is aligned with findings from another study that focused on the adoption of smart home among the elderly in KSA [33]. Providing adequate training to elderly people with low education levels might encourage them to try and practice using mHealth technology. Differences among females and males regarding perceived usefulness, perceived ease of use, and attitudes were clear in the findings. Similarly, a recent study found that perceived usefulness and attitude were more strongly linked to intention to use for males than for females. However, perceived ease of use was not different between males and females [14].

Prior studies have examined the intention to use mHealth among the elderly using several models in different countries. A study based on the UTAUT model in Bangladesh found that performance expectancy, effort expectancy, social influence, technology anxiety, and resistance to change had a significant impact on the elderly’s intention to use mHealth services [31]. Their results are in line with our results only in the perceived ease of use factor, which is directly related to effort expectancy. A study conducted in Pakistan based on the UTAUT theoretical framework showed that performance expectancy, effort expectancy, social influence, and facilitating conditions have a positive significant relationship with the intention to use mHealth among the elderly [37]. Our results align with their results in effort expectancy and facilitating conditions in terms of family support. However, their findings did not specify the type of facilitating condition available. A recent study based on the decomposed theory of planned behavior conducted in China suggested that knowing about mobile health, usage of mobile health, self-rated health, perceived ease of use, perceived usefulness, and self-efficacy were important factors influencing the willingness to use mHealth among elderly [15]. Their findings are similar to our results regarding perceived ease of use, perceived usefulness, and familiarity. Although, the population in these studies is elderly. However, using different theoretical models has a significant impact on the studies’ findings. This study’s findings are based on the H-TAM model which provides a structured framework for comprehending how the elderly embrace and utilize technology in the healthcare context. Furthermore, populations in these studies are from different countries with heterogeneous social and health factors which might explain the variation in results among these studies.

This study has several limitations. Firstly, sample selection, primarily through researcher contacts and retirement associations, may introduce sampling bias, potentially limiting the generalizability of the findings. However, the participants were heterogeneous in gender, education level, and age. Moreover, most of our participants were male, married, and had chronic diseases, which is similar to reported elderly statistical data in Saudi Arabia, according to the General Authority for Statistics [38,39]. Our sample demographic characteristics were comparable to other studies’ participants in term of gender, marital status, and having chronic disease [14,31,37]. Three participants were younger than the other participants (aged more than 50 years old). Despite the age difference, the younger participants had fewer positive attitudes toward mHealth than other participants, and their educational levels and technological familiarity were varied, which might not have had an impact on the study’s findings. Additionally, self-reported data collected through interviews may be susceptible to recall bias, affecting response accuracy. Lastly, the number of the sample was 14 participants, which might be considered small for drawing generalizable conclusions. However, it is considered within the acceptable number of participants in qualitative studies according to Hennink and Kaiser [40]. More importantly, the author conducted two interviews after saturation was reached to confirm that the situation was maintained [41].

Future research should improve our understanding of mHealth technology acceptance among older adults. Longitudinal studies can track intentions and behaviors over time, providing a comprehensive view of the adoption process. Intervention studies are needed to evaluate targeted programs aimed at enhancing acceptance, including digital literacy initiatives and educational campaigns. Furthermore, comparative studies across countries can illuminate how demographic and social factors influence mHealth technology acceptance in diverse populations.

## 5. Conclusions

This qualitative study explains the factors influencing intentions to use mHealth and identifies barriers and suggested solutions among elderly people aged 50+ years in KSA guided by H-TAM. H-TAM provides a comprehensive framework for understanding the intention to use mHealth among older adults. By analyzing interview responses through this lens, we recognize the multifaceted nature of factors influencing the intention to use mHealth, encompassing age, education, gender, perceived usefulness, perceived need, perceived ease of use, perceived advantages, familiarity, trust in technology, advice acceptance, support, and compatibility factors. Several central barriers emerged, including lack of digital literacy, health and aging issues, worry about making mistakes, and social issues. Tailored educational and training campaigns, user-friendly interfaces, and simplicity in mHealth apps’ design can mitigate these barriers. Addressing these factors in the design, promotion, and support of mHealth can promote their successful adoption among older populations. Future work is still needed to illustrate how demographic and social factors influence mHealth acceptance in diverse populations.

## Figures and Tables

**Figure 1 healthcare-11-03024-f001:**
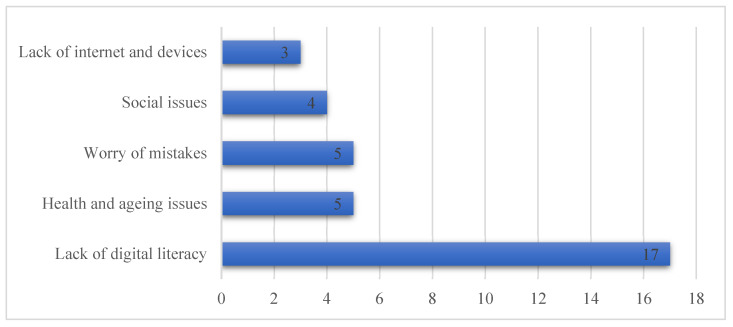
Frequency of barriers affecting the intention to use mHealth among the elderly.

**Figure 2 healthcare-11-03024-f002:**
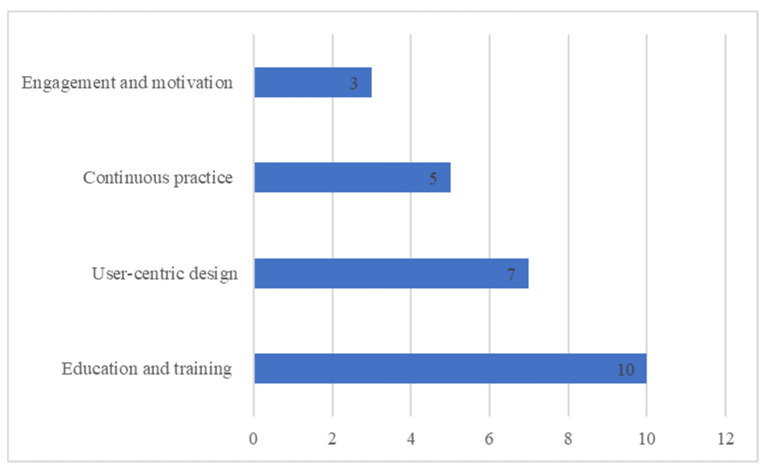
Frequency of solutions and recommendations to overcome barriers related to the intention to use mHealth among the elderly.

**Table 1 healthcare-11-03024-t001:** Demographic data of participants.

Characteristic	N	(%)	Mean (SD)
Age			62.6 (5.9)
Gender			
Male	8	57.2	
Female	6	42.8	
Education			
Middle school graduate	3	21.43	
High school graduate	3	21.43	
Bachelor’s Degree	4	28.57	
Master’s Degree	1	7.14	
Ph.D. Degree	3	21.42	
Marital status			
Married	12	85.71	
Widowed	2	14.29	
Income per month			
0–10,000	6	42.86	
>10,000–20,000	5	35.71	
>20,000 and more	3	21.43	
Self-Rated Health			
Fair	1	7.143	
Good	6	42.86	
Very good	4	28.57	
Excellent	3	21.43	
Chronic Diseases			
Yes	10	71.43	
No	4	28.57	

**Table 2 healthcare-11-03024-t002:** Factors affecting intention to use mHealth among the elderly based on H-TAM model.

Factor	Type	Count	Quotes from Participants
**Sub-theme 1**			
Role of demographics and health status	Age	12	“*Especially those over 80 years old, they will not use it*”.
	Education	16	“*The more educated an older person is, the easier the technology will be for him*”.
	Health status	13	“*It depends on the extent to which the person cares about his health. There is a difference between a person who is concerned about his health, and exercise, and a person who is not concerned*”.
	Gender	14	“*I prefer to go to the hospital and book appointments. As for the app, I don’t like to use it*”.
**Sub-theme 2**			
Perceived Usefulness	Useful	12	“*Yes, it is very useful because health awareness has become widespread among the elderly*”.
	Not useful	5	“*I prefer to go to the hospital and book appointments. As for the technology, I don’t like to use it…*”
Perceived Benefit		30	“*It saves time instead of going to the hospital to make appointments. Also, if I need medical information, I can contact the virtual consultation service and ask them instead of going to the doctor*”.
Perceived Need	For oneself	12	“*Yes, I need it. For example, book appointment*”
	For others	11	“*Most people are using mobile phones*”.
Relative Advantage	Not using similar technology	5	“*No, I don’t have any similar technology*”.
	Using similar technology	4	“*Yes, there is a special technology [app] for the university hospital, but it is difficult to book appointments because they are only available within 3 months*”.
**Sub-theme 3**			
Perceived Ease of Use	Easy	9	“*It’s easy, just click on the link*”.
	Not easy	6	“*But if he is uneducated, uncultured, and does not care about his health, it will be difficult for him*”.
Convenience	Convenience	8	“*There is no effort or fatigue*”.
	Inconvenience	6	“*Make a greater effort to learn it and use it correctly*”
Familiarity	Yes	9	“*Yes, I know about this technology*”.
	No	2	“*I don’t know it*”
Privacy	Invades privacy	8	“*The topic is sensitive. No one wants to reveal that he has such-and-such a disease because they do not want anyone to know about the disease, or their medical history, as it is confidential information*”.
	Lack of privacy concern/not an issue	11	“*There is no concern because there is no financial or cognitive return from hacking it*”.
Trust	In technology	11	“*I definitely trust it*”.
	In person	2	“*Because those responsible for it are trustworthy people*”.
	Lack of trust	1	“*I’d rather go to the hospital*”.
**Sub-theme 4**			
Subjective Norm Health care provider	Positive	4	“*He’ll see that I’m a advanced person with the ability to use mHealth technology*”.
	Not care	6	“*I don’t know*”
Subjective Norm Family	Positive	5	“*My family is also happy that I use these technologies and rely on myself*”.
	Not care	3	“*This is up to me. There is no one who encouraged me or discouraged me from using mHealth technology*”.
Subjective Norm Friends	Positive	5	“*…I expect that they will be happy…*”
	Not care	1	“*I don’t know*”
Advice Acceptance Healthcare provider	Positive	11	“*…if I do not have it, I should ask the doctor for information about the technology and how to use it…*”
	Negative	2	“*No, because I have many mHealth applications…*”
Advice Acceptance Family	Positive	10	“*Yes, it excites me*”.
	Negative	3	“*No, I am satisfied with the mHealth applications on my phone*”.
Advice Acceptance Friends	Positive	6	“*Of course, if I know its benefits, I will help and spread it*”.
	Negative	2	“*I do not trust people from outside my family*”.
**Sub-theme 5**			
Compatibility	Compatible	8	“*Yes, it is easy to integrate*”.
	Incompatible	4	“*No, on a daily basis, it will be difficult, because it takes time*”.
Price Value	Pay	8	“*If I am used to it and comfortable using it, I will pay*”.
	Not pay	7	“*No, I will not pay*”.
Support availability	Family support	19	“*First I need someone from my family to teach me and then I will be able to use it*”.
	No support	8	“*I rely on myself and am not intruding on others…*”
**Sub-theme 6**			
Attitudes	Positive	9	“*I feel it is a safety valve for those who understand, and those who learn begin to benefit from them after*”.
	Negative	5	“*But I want to go to the hospital, it is better for me.* *Because I don’t like using the mobile phone a lot, only messages or calls*”.

## Data Availability

Data are contained within the article.

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
