# Peer review of "Factors, Barriers, and Recommendations Related to Mobile Health Acceptance among the Elderly in Saudi Arabia: A Qualitative Study"

_healthcare, 2023, doi:10.3390/healthcare11233024_

Round 1
Reviewer 1 Report
Comments and Suggestions for Authors
Thank you for the opportunity to review this paper. It presents an interesting and important study that explores the factors influencing the intention to accept or reject the use of e-health technologies among the elderly. Please find my comments below:
Introduction:
The authors might consider incorporating some previous related studies for comparison, highlighting the differences between them.
Materials and Methods:
Introducing subheadings such as "Participants" and "Data Collection" and numbering them as 2.1, 2.2, etc., could enhance the structure of this section.
Incorporating a study framework would be beneficial to help readers understand the study concept more easily.
Results:
I echo my previous suggestion: visualizations could help readers grasp the main points more quickly.
Discussion:
The authors might consider providing specific suggestions for future research or policy-making.
Reviewer 2 Report
Comments and Suggestions for Authors
Thank you for entrusting me with the review of this manuscript, which investigates the factors impacting the adoption of e-Health technology by the elderly population in Saudi Arabia. Overall, the topic of the manuscript and the methodological approach are current and of interest. The following are my comments:
-The term "e-health technology," as used in the manuscript, is somewhat misleading given that the interview questions in Appendix A are specific to the Sehhaty app. To better align with the scope of your study as evidenced by the interview content, I recommend refining the term to "mobile health." This narrower focus should be introduced early in the manuscript, with adjustments made to the introduction to reflect this change.
-Regarding the abstract, it is essential to follow the journal's guidelines for structuring abstracts. Please consult the Instructions for Authors (https://www.mdpi.com/journal/healthcare/instructions) to ensure compliance with the abstract format and word count requirements.
-In the Background section (line 6), the research question should be explicitly stated. This inclusion will align with the editorial advice provided in the Instructions for Authors, which is “Place the question addressed in a broad context and highlight the purpose of the study.”
-In the Conclusions section (line 18), ensure that the conclusions drawn are in sync with the stated objectives of the study.
-The Results section delineates three themes, yet the title and objectives seem to address only the first. To resolve this discrepancy, consider revising the title, objectives, and corresponding sections of the manuscript to encapsulate all three identified themes.
-Line 62. “However, no studies have qualitatively discussed factors influencing the intention to use e-health technologies among the elderly.” I would recommend adding "in Saudi Arabia".
- Line 112. “As a result of our analyses, we assigned 81 codes and created 3 major themes.” It is more suitably placed in section 3.2, under Thematic Analysis.
-I advise restructuring Table 1 to succinctly present demographic characteristics using statistical measures such as mean (standard deviation), counts, or percentages, rather than individual participant data.
- Line 130. ”Sub-theme 1: role of demographics and health status” is not in Table 2.
-For Table 2, the inclusion of 'Sub-theme' in the first column would provide clarity and consistency within the table.
-On page 8, ensure that all quotations are properly closed. For instance, the phrase "Most people are using mobile phones" lacks a closing quotation mark.
-On page 8, "Lack of privacy". The count and Quotes from participants are missing.
-On page 8. “I don't know” lacks an opening quotation mark.
-Ensure the correct use of quotation marks throughout the manuscript. For example, there should be no space between the opening quotation mark and the first word of the quote (line 323), and punctuation marks should be placed inside the closing quotation marks, unless followed by a citation (line 249).
line 323 “ Health challenges, such as poor eyesight, may be the reason that prevents you from using the technology.”.
line 249 “It is normal and the doctor will not ask about it.”.
-Line 369. “at first it will be difficult, but if you get used to it, it will become easier” I would suggest using italics for emphasis and consistency throughout the manuscript.
-There is a need to rectify the numbering of the tables; specifically, Table 3(line 338) should be renumbered as Table 4, and a new Table 3 should be introduced under the section "Theme 2: Concerns and Barriers."
-In the Conclusions section, I encourage a strong reiteration of the study’s objectives and a discussion that stays directly pertinent to those aims.
-The manuscript would be strengthened by engaging with existing research on investigating the barriers to effective diabetes self-management in older populations when using mobile applications. Notably, Ye et al. (2018) provide a pertinent analysis of the challenges faced by older people, detailing specific barriers that impede effective self-management through mobile technology. Incorporating this study into the manuscript's references would provide a more complete picture of older adults' user experiences with mobile health technologies.
Ye, Q., Boren, S. A., Khan, U., Simoes, E. J., & Kim, M. S. (2018). Experience of diabetes self-management with mobile applications: a focus group study among older people with diabetes. European Journal for Person Centered Healthcare, 6(2), 262-273.
Comments on the Quality of English LanguageThe manuscript is well-written, and the quality of the English language is commendable.
Reviewer 3 Report
Comments and Suggestions for Authors
Dear Authors,
I have reviewed the manuscript “Factors Influencing E-Health Technology Acceptance Among the Elderly In Saudi Arabia: Qualitative Study” Manuscript ID: healthcare-2711415 that has been submitted for publication in the: Healthcare (ISSN 2227-9032), and I have identified a series of aspects that in my opinion must be addressed in order to bring a benefit to the manuscript.
The article under review will be improved if the authors address the following aspects in the text of the manuscript:
1. In the abstract, the sample size of 14 older adults may be considered small for drawing generalizable conclusions, particularly in a diverse population like the elderly aged 50+ in Saudi Arabia. The study could benefit from a more extensive and diverse sample to ensure broader insights into the factors influencing e-health technology acceptance.
2. While thematic content analysis is a valuable qualitative method, the abstract could benefit from a brief explanation of how rigor and reliability were maintained in the analysis process to enhance the trustworthiness of the results.
3. The abstract mentions age and level of education as significant moderators. It would be helpful to elaborate on how these variables were specifically found to moderate the intention to use e-health technology among the elderly.
4. Compare the research methodologies employed by different studies. Assess the strengths and weaknesses of each approach, including sampling methods, data collection techniques, and data analysis procedures.
5. Compare the characteristics of the study samples, such as age range, gender distribution, and any other relevant demographic factors. Differences in sample characteristics can impact the generalizability of findings.
6. Examine the specific variables or factors investigated in each study. Identify commonalities and differences in the factors influencing e-health technology acceptance among the elderly.
7. The references need to be updated for the years 2022 and 2023, as this field has been recently raised.
https://doi.org/10.1016/j.engappai.2023.107261
8. The authors should provide more details in the conclusion about the future work that will be done later.
Round 2
Reviewer 2 Report
Comments and Suggestions for Authors
The authors have satisfactorily addressed all my concerns in my previous reviews. The revisions have notably improved the manuscript. However, I have a few additional suggestions that could enhance clarity and accuracy:
-Line 111. Please verify the appropriateness of the citation “(p.450)” in this context. It seems potentially out of place or incorrectly formatted.
-Table 1, Age section. I would suggest aligning the “Age” characteristic to the left, consistent with the formatting of other characteristics in Table 1. Additionally, it would be beneficial to present the mean and standard deviation for age in a separate column. This could be formatted as “Mean (SD)”, for example, “62.6 (5.9)”, to enhance clarity and ease of interpretation for the readers."
-Table 1, Education section. I noticed a numerical inconsistency. The sum of participants across the various education categories amounts to only 11, which contrasts with the expected total of 14. I recommend a thorough review of these figures to ensure accuracy in the data representation.
Author Response
The author thanks the reviewers for their insightful critiques of our manuscript. We have carefully reviewed all the comments and addressed them with revisions, highlighted in the revised manuscript. We believe the manuscript has significantly improved and hope it will now be suitable for publication.
-Line 111. Please verify the appropriateness of the citation “(p.450)” in this context. It seems potentially out of place or incorrectly formatted.
The citation has been corrected.
-Table 1, Age section. I would suggest aligning the “Age” characteristic to the left, consistent with the formatting of other characteristics in Table 1. Additionally, it would be beneficial to present the mean and standard deviation for age in a separate column. This could be formatted as “Mean (SD)”, for example, “62.6 (5.9)”, to enhance clarity and ease of interpretation for the readers."
The age section has been aligned to the left and presented as mean and SD in a separate column.
-Table 1, Education section. I noticed a numerical inconsistency. The sum of participants across the various education categories amounts to only 11, which contrasts with the expected total of 14. I recommend a thorough review of these figures to ensure accuracy in the data representation.
I apologize for this mistake. A fourth category has been added. All tables and figures have been revised.
Reviewer 3 Report
Comments and Suggestions for Authors
Accept in present form
Author Response
The author thanks the reviewer for their insightful critiques of our manuscript. We believe the manuscript has significantly improved and hope it will now be suitable for publication.